# Differential Pharmacokinetics of Liver Tropism for Iron Sucrose, Ferric Carboxymaltose, and Iron Isomaltoside: A Clue to Their Safety for Dialysis Patients

**DOI:** 10.3390/pharmaceutics14071408

**Published:** 2022-07-05

**Authors:** Guy Rostoker, Fanny Lepeytre, Myriam Merzoug, Mireille Griuncelli, Christelle Loridon, Ghada Boulahia, Yves Cohen

**Affiliations:** 1Division of Nephrology and Dialysis, Ramsay Santé, Hôpital Privé Claude Galien, 91480 Quincy-Sous-Sénart, France; fanny.lepeytre@gmail.com (F.L.); myriam.merzoug@ramsaysante.fr (M.M.); mgriuncelli@free.fr (M.G.); loridon@free.fr (C.L.); boulahiaghada@yahoo.fr (G.B.); 2Collége de Médecine des Hôpitaux de Paris, 75005 Paris, France; 3Division of Radiology, Ramsay Santé, Hôpital Privé Claude Galien, 91480 Quincy-Sous-Sénart, France; yvescohen.imagerie@gmail.com

**Keywords:** pharmacokinetics, biodistribution, intravenous iron, iron sucrose, ferric carboxymaltose, iron isomaltoside, magnetic resonance imaging (MRI), end-stage kidney disease (ESKD), iron overload, liver iron concentration (LIC)

## Abstract

Anemia is a major complication of end-stage kidney disease (ESKD). Erythropoiesis-stimulating agents and intravenous (IV) iron are the current backbone of anemia treatment in ESKD. Iron overload induced by IV iron is a potential clinical problem in dialysis patients. We compared the pharmacokinetics of liver accumulation of iron sucrose, currently used worldwide, with two third-generation IV irons (ferric carboxymaltose and iron isomaltoside). We hypothesized that better pharmacokinetics of newer irons could improve the safety of anemia management in ESKD. Liver iron concentration (LIC) was analyzed in 54 dialysis patients by magnetic resonance imaging under different modalities of iron therapy. LIC increased significantly in patients treated with 1.2 g or 2.4 g IV iron sucrose (*p* < 0.001, Wilcoxon test), whereas no significant increase was observed in patients treated with ferric carboxymaltose or iron isomaltoside (*p* > 0.05, Wilcoxon-test). Absolute differences in LIC reached 25 μmol/g in the 1.2 g iron sucrose group compared with only 5 μmol/g in the 1 g ferric carboxymaltose and 1 g iron isomaltoside groups (*p* < 0.0001, Kruskal–Wallis test). These results suggest the beneficial consequences of using ferric carboxymaltose or iron isomaltoside on liver structure in ESKD due to their pharmacokinetic ability to minimize iron overload.

## 1. Introduction

Iron is a fundamental element implicated in DNA synthesis and repair and hemoglobin and myoglobin composition, and is tightly regulated at the cellular level to avoid toxicity of free iron and its consequence, cellular death mediated by ferroptosis [1,2,3]. Iron metabolism in mammals and humans is a closed system, regulated by the master hormone hepcidin locking the entry of iron in the organism by regulating ferroportin, the sole known iron-export cellular protein; hepcidin production by the liver is antagonized by a second iron hormone, erythroferrone, produced by medullary erythroblasts in response to hemorrhage and erythropoietin [4].

Anemia is a major complication of end-stage kidney disease (ESKD). The discovery of epoetin 30 years ago enabled the avoidance of blood transfusions and their drawback, human leucocyte antigen (HLA) sensitization, improving clinical outcomes and quality of life for ESKD patients [5,6]. Erythropoiesis-stimulating agents (ESA) and intravenous (IV) iron are the current backbone of anemia treatment in ESKD, and almost all hemodialysis patients (about 4 million worldwide) receive IV iron to compensate for iron deficiency due to significant blood loss during dialysis, to ensure efficient iron-dependent erythropoiesis with ESA, and to overcome ESKD functional and true iron deficiency [7,8,9].

In ESKD, IV iron has been shown to be superior to oral preparations (in both hemodialysis and peritoneal dialysis), which are poorly tolerated, to allow cost savings of about 25% on expensive ESA molecules, and to decrease hospitalization rates related to cardiac insufficiency, as demonstrated in the recent PIVOTAL trial [9,10].

In the current ESA era, iron overload in ESKD is a matter of debate [11,12]. However, it is now observed increasingly by quantitative magnetic resonance imaging (qMRI) in up to 66% of hemodialysis patients treated with second-generation IV irons, namely iron sucrose, iron gluconate, and iron polymaltose [13,14,15], and a recent autopsy study in Portuguese dialysis patients found similar results (55%) [16]. Liver iron accumulation in dialysis patients increases hepcidin production [13,14], which has been associated with a risk of cardiovascular events and mortality [17,18]. Moreover, liver iron accumulation in dialysis patients has recently been shown to increase liver fat fraction, with the ability to induce or worsen fatty liver disease [19], and qMRI has recently been shown to non-invasively and reliably assess the biodistribution of iron sucrose in rodents [20].

Taking into consideration the fact that ESA and IV iron are the gold-standard treatment for ESKD anemia and will remain so for the considerable future, and in the view of the recent refusal of marketing authorization (MA) for roxadustat (the first of the new pharmacological class of hypoxia-inducible factor (HIF)-stabilizers) by the Food and Drug Administration in the USA [21], together with restriction of its MA by the European Medicines Agency in the EU [22], we hypothesized that a pharmacokinetic study of liver accumulation of new third-generation IV irons (namely, ferric carboxymaltose and iron isomaltoside) compared with that of iron sucrose, the main cause of hemodialysis-associated hemosiderosis in radiological studies [13,14,15], could provide data on the safety of anemia management in ESKD. 

## 2. Materials and Methods

### 2.1. Study Design and Patients

This ancillary, observational, longitudinal study was carried out between 7 August 2013 and 14 January 2020, at Claude Galien Hospital (Quincy-sous-Sénart, France). All participants gave their written informed consent. The inclusion and exclusion criteria of this study have been described previously [14].

This study received technical and ethical approval from the Drug, Devices, and Clinical Trials Committee of our institution (COMEDIMS Claude Galien, 9 December 2004, and 15 February 2013) and was conducted in accordance with the Declaration of Helsinki.

This study is registered under International Standard Randomized Controlled Trial Number (ISRCTN) 80100088 [23]; of note, ISRCTN registers both clinical trials and observational cohort studies. A compliance commitment MR-4 was declared to the devoted Computing French Commission (CNIL). 

### 2.2. Anemia Treatment and Iron Therapy 

The anemia treatment used in this study has been described previously [14]. Briefly, the patients were treated for anemia according to European best practice guidelines with a reactive strategy of iron store replenishment [8]. The treatment of anemia in these patients remained unchanged during the study; it comprised, when required, ESA in hemodialysis and peritoneal dialysis patients, and iron therapy. For IV iron in hemodialysis patients, iron sucrose (Mylan^®^, Saint-Priest, France) was used in most patients and third-generation IV irons (namely ferric carboxymaltose (Ferinject^®^, Vifor Pharma, Paris La Défense, France) and iron isomaltoside (Monover^®^, Aguettant, Lyon, France)) in a few patients [8,9].

With the aim of minimizing the risk of liver iron overload in our hemodialysis patients in view of our previous publication [14], iron replenishment comprised, according to the French label [24], a total infusion of 1.2 g of iron sucrose over a period of 3 months (100 mg/week given at a mid-week dialysis session, with an infusion of 2 h duration beginning after 1 h of dialysis) in patients with moderate iron deficiency, and a total infusion of 2.4 g of iron sucrose over a period of 6 months (100 mg/week given at mid-week dialysis session with an infusion of 2 h duration beginning after 1 h of dialysis) in patients with severe iron deficiency. Ferric carboxymaltose and iron isomaltoside were given to hemodialysis patients with moderate iron deficiency, according to their respective French and European labels, as a total dose of 1 g (200 mg during five successive dialysis sessions for ferric carboxymaltose and 500 mg during two successive dialysis sessions for iron isomaltoside, with an infusion of 2 h duration beginning after 1 h of dialysis) [25,26]. 

Oral iron was used as first-line therapy in peritoneal dialysis patients and third-generation IV irons (ferric carboxymaltose and iron isomaltoside) were used in these patients as second-line treatment in the case of intolerance to oral compounds, severe iron deficiency, or iron deficiency unresponsive to oral iron therapy [8]. Ferric carboxymaltose and iron isomaltoside were given to peritoneal dialysis patients as a single 1 g infusion (over a duration of 2 h followed by a 30-min follow-up for safety reasons in the dialysis center) according to their respective French and European labels [25,26].

### 2.3. Longitudinal Analysis of Liver Iron and Spleen Iron Concentrations 

Changes in liver iron concentration (LIC) and spleen iron concentration (SIC) were closely monitored during iron therapy by repeated quantitative hepatic and splenic MRI performed before and after iron therapy. 

### 2.4. Biological Markers 

The efficacy of anemia treatment was determined routinely using a hemoglobin assay and reticulocyte counts every 2 weeks in patients on hemodialysis and monthly in those treated by peritoneal dialysis. Monthly measurements of iron biomarkers (ferritin, transferrin, serum iron, transferrin saturation, hemoglobin reticulocyte content) and C-reactive protein were also performed routinely in both hemodialysis and peritoneal dialysis patients. Initial biological markers were measured as close as possible to the first MRI before initiation of IV iron infusion, and final biological markers were measured 2 months after the completion of iron therapy.

### 2.5. Quantification of LIC and SIC by MRI 

MRI measurements were all taken by the same senior radiologist (Y.C.) who was unaware of the patients’ medical history (with the exception of their dialysis method), modality of IV iron therapy, and iron biomarker values for follow-up of anemia treatment. An Optima^TM^ MR450 MRI machine was used until 2018 and thereafter a SIGNA^TM^ Artist MRI machine (GE Medical Systems, Milwaukee, WI, USA) was used, both operating at a field strength of 1.5 Tesla. 

Of note, quantitative liver MRI has been performed routinely at the radiology division of our hospital since 2005 and has been used in ordinary care since this date for the follow-up of iron stores of dialysis patients at Claude Galien hospital [14]. Wherever possible, patients on iron therapy received their last iron dose at least 1 week before MRI. During the MRI session, LIC was assessed by signal intensity ratio (SIR) according to Rennes University and R2* relaxometry was used to determine SIC. 

*LIC*: The method used for measurement of LIC was based on T1 and T2* contrast imaging (without gadolinium), as established by Gandon et al. at Rennes University in 2004 and validated in 191 patients with genetic hemochromatosis and secondary hemosiderosis who underwent liver biopsy for biochemical iron assay [27]. This method has recently been shown to accurately measure iron load in hemodialysis patients when compared with quantitative liver histology (Deugnier and Turlin Scoring) with Perls staining [28]. Free analytical software was available on the Rennes University website and LIC was expressed in μmol/g dry liver [27]. Normal LIC values were set at <40 μmol/g dry weight according to Rennes University and our previous work [27,28].

*SIC*: Spleen iron concentration was assessed by R2* relaxometry according to Wood et al. [29] and expressed as R2* in Hz and converted into T2* in ms. SIC was extrapolated in μmol/g using the Garbowski equation (used primarily in R2* relaxometry to translate hepatic T2* expressed in ms into μmol/g dry liver) [30]. 

Of note, in France, quantitative hepatic MRI has been fully reimbursed by the universal national health insurance system (Sécurité Sociale) for the diagnosis and monitoring of iron overload diseases (a cost of around €350 including radiologist fees) since 2005, and was advocated in 2015 for the follow-up of iron stores in French dialysis patients [7].

### 2.6. Statistical Analyses 

As our data did not conform to a Gaussian distribution (Shapiro–Wilk normality test) all data are expressed as median and range (min–max) [31]. Categorial data are given as percentage (%). The different groups of patients were compared using non-parametric analysis of variance with the Kruskal–Wallis test for continuous variables, followed by Dunn’s post-test, and with the Chi^2^ test for categorical variables [31].

As the data obtained at the two time points in patients under different IV iron therapies did not conform to a Gaussian distribution (Shapiro–Wilk normality test), values for LIC and SIC in each group of patients were analyzed using the non-parametric paired two-tailed Wilcoxon test [31].

For analysis of the percentage of total IV iron sequestered in the liver and spleen in patients receiving 1.2 g iron sucrose, 2.4 g iron sucrose, 1 g ferric carboxymaltose, or 1 g iron isomaltoside, we added up LIC and SIC in μmol/g in each group and extrapolated the quantity in g by using the Brissot equation established by an initial liver biopsy in genetic hemochromatosis patients treated by phlebotomy (130 μmol LIC = 1 g of iron) [32]. The ratio of sequestered iron (g)/iron infused (g) for the three different iron compounds was then compared using the Chi^2^ test [31]. PRISM 9 software (GraphPad, San Diego, CA, USA) was used for all statistical analyses and *p* < 0.05 was considered to be statistically significant [31].

## 3. Results

### 3.1. Characteristics of the Study Population 

The study cohort comprised 54 adult ESKD patients: 47 treated by intermittent hemodialysis and 7 by peritoneal dialysis.

There were 11 patients who received 1.2 g iron sucrose (group A), and 14 patients who received 2.4 g iron sucrose (group B); 15 patients were treated with 1 g ferric carboxymaltose (group C) and 14 received 1 g iron isomaltoside (group D). Before this cycle of IV iron therapy, 6/54 patients (11.1%) had abnormal LIC (defined as ≥40 μmol/g); one in group A, two in group B, two in group C, and one in group D.

A total of 74 patients were initially screened; 20 patients were excluded from the analysis for the following reasons: lack of a second qMRI (*n* = 5), second qMRI performed too late after the end of iron therapy (*n* = 9), lack of infusion of the full dose of third-generation IV iron (*n* = 2), patient receiving excessive iron sucrose dose (3 g) (*n* = 1), patient receiving both iron isomaltoside and iron sucrose during the study period (*n* = 1), genetic hemochromatosis (*n* = 1), and chronic severe inflammatory process occurring during the study (*n* = 1).

The median period of time between the first qMRI and initiation of IV therapy was 22 days (range: 0–162) and the last qMRI was performed a median of 44.5 days (range: 6–69) after the completion of iron treatment. The demographic and clinical characteristics of these patients are summarized in Table 1.

### 3.2. Evolution of Liver Iron Load by MRI 

LIC was similar in the four groups of patients before iron therapy (*p* = 0.531, Kruskal–Wallis test). Overall, after their cycle of IV iron therapy, 23/54 patients (42.6%) had abnormal LIC; seven in group A, thirteen in group B, two in group C, and one in group D. The increase in the number of patients with abnormal LIC under iron therapy was only observed in group A (from one to seven patients) and in group B (from two to thirteen patients). LIC after IV iron therapy increased significantly in group A (iron sucrose 1.2 g; *p* = 0.001, Wilcoxon test) and group B (iron sucrose 2.4 g; *p* = 0.0001, Wilcoxon test) whereas there was no significant difference in LIC in group C (ferric carboxymaltose) or group D (iron isomaltoside) (*p* > 0.05, Wilcoxon test) (Table 2 and Figure 1). 

The absolute difference in LIC (measured as LIC after iron therapy minus LIC before IV iron) also differed strikingly between iron sucrose (25 μmol/g in group A (1.2 g) and 35 μmol/g in group B (2.4 g)) compared with ferric carboxymaltose (group C) and iron isomaltoside (group D) (5 and 5.5 μmol/g, respectively; *p* < 0.0001, Kruskal–Wallis test) (Figure 2).

### 3.3. Evolution of Splenic Iron Load by MRI 

SIC by qMRI was similar before iron therapy in the four groups of patients (*p* = 0.799, Kruskal–Wallis test). The increase in SIC after iron therapy was less than the increase in LIC and only concerned patients in groups A and D (Table 2 and Figure 3). Similarly, the absolute difference in SIC (measured as SIC after iron therapy minus SIC before IV iron) was smaller in magnitude than that observed for LIC (Table 2 and Figure 2). 

### 3.4. Quantification of Iron Sequestered in the Liver and Spleen after IV Iron Infusions 

The percentage of iron perfused and sequestered in the reticuloendothelial system (e.g., liver plus spleen) differed significantly between iron sucrose 1.2 g and third-generation irons (infusions of 1 g): this increased to 21.6% in group A (iron sucrose 1.2 g) compared with 5.6% in group C (ferric carboxymaltose 1 g), and 8.6% in group D (iron isomaltoside 1 g) (*p* = 0.0012, Chi^2^ test) (Figure 4). The main driver of this phenomenon was liver iron storage (*p* = 0.0015, Chi^2^ test), but not splenic iron storage (*p* = 0.352, Chi^2^ test) (Figure 4). Of note, the percentage of iron perfused and sequestered in the reticuloendothelial system for group B (2.4 g of iron sucrose infused over 6 months) reached 12.7% (Figure 4).

### 3.5. Evolution of Iron Biomarkers 

The biological characteristics of the patients during the study are summarized in Table 2. Overall, hemoglobin and biological markers showed similar trends in the four IV iron therapy groups (Table 2).

## 4. Discussion

In this qMRI study in dialysis patients, significant differences in liver tropism were observed between widely used iron sucrose and two third-generation IV irons, namely ferric carboxymaltose and iron isomaltoside, providing an important indication of their potential safety in dialysis patients. Although the increase in SIC after iron therapy was less than the increase in LIC, the splenic iron increase was minimal with ferric carboxymaltose, whereas iron isomaltoside had a superior increase in the spleen but this was lower than that with 1.2 g iron sucrose (Table 2 and Figure 4) suggesting a difference in splenic storage between these molecules. 

The liver is the major site of iron storage in humans and LIC has been shown to closely correlate with total iron stores both in patients with genetic hemochromatosis and in those with secondary hemosiderosis [33,34]. In the last decade, qMRI has become the gold-standard method for LIC quantification and non-invasive follow-up of patients with iron overload disorders [33,34]. Concomitantly, this modern imaging technique has demonstrated a high frequency of liver iron overload in hemodialysis patients, affecting about two-thirds of patients, with potential cardiovascular events due to iatrogenic hyperhepcidinemia and potential iatrogenic fatty liver disease [15,17,19]. Radiological iron overload in dialysis patients has been described with second-generation IV irons, namely iron sucrose (either the original Venofer^®^ (two studies comprising 140 patients) or its generics (three studies comprising 184 patients)), iron polymaltose (Ferrosig^®^; two Australian studies totalling 25 patients), and iron gluconate (Ferrlecit^®^; 40 patients) [15].

The findings of this study on liver accumulation of iron sucrose are similar to those observed in an Australian study with iron polymaltose (Ferrosig^®^) given at a dosage of 1–1.5 g to 25 iron-deficient non-dialysis patients with chronic kidney disease stages 3–5, where R2 relaxometry MRI showed an increase of 25.4 μmol/g 2 weeks after the infusion [35]. Our results are also in line with the rapid transient increase in exchangeable compartments of iron, especially in the liver and spleen, in the few days after a single infusion of iron sucrose described by Beshara et al. using positron emission tomography, compared with weak tropism for the liver with iron carboxymaltose [36,37,38].

Our academic, non-industry driven study provides additional answers on the safety of IV iron products to the question of the Pharmacovigilance Committee of European Medicines Agency in September 2017, stipulating that pharmaceutical companies with MA for iron products should “Investigate the risk of iron overload, particularly in chronic kidney disease (CKD) patients and in patients with inflammatory bowel disease (IBD) and provide a cumulative review of all cases of iron overload reported with iron-containing products (verbatim)” [38]. 

We feel that our findings on the potential differential liver tropism of IV iron compounds may override the apparent contradictory data on iron therapy in non-dialysis CKD patients, indicating high safety of ferric carboxymaltose compared with toxicity and death due to cardiovascular and infectious causes with iron sucrose [9]. Furthermore, avoidance of the development or aggravation of fatty liver disease with ferric carboxymaltose and iron isomaltoside might be of value in overweight patients (20–30% of hemodialysis patients in Western countries, more in the USA), patients with known non-alcoholic fatty liver disease, and those with dysmetabolic iron overload syndrome [39].

Finally, these two third-generation IV irons may also be useful in dialysis patients suffering from hepatitis C and B where iron can aggravate the liver disease [40].

Nevertheless, this study has four main limitations: first, the iron sucrose doses were unusually spread out over time compared with current practice (with the aim of reducing the risk of liver iron overload) and therefore probably minimized the increase in LIC and SIC. Second, the equation for calculating SIC was initially devoted to LIC and was extrapolated from it, therefore giving a potential approximation in the results. Third, the study was observational, in a daily practice setting, explaining variations in the delay to the final MRI, which may have influenced the LIC and SIC results with a potential trend to minimization. Fourth, the observational nature of this ancillary study, without randomization of patients to the different IV iron treatments, as well as the likely different ESKD trajectories and comorbidities, could have influenced the results of LIC and SIC independently of the drugs. 

## 5. Conclusions

The potential beneficial consequences of ferric carboxymaltose and iron isomaltoside on liver structure, with suggested weak accumulation compared with that of iron sucrose, warrant a rigorous demonstration by prospective clinical trials. Potential weak accumulation of ferric carboxymaltose and iron isomaltoside in the liver would also reinforce their safety in IBD and cardiac insufficiency. 

## Figures and Tables

**Figure 1 pharmaceutics-14-01408-f001:**
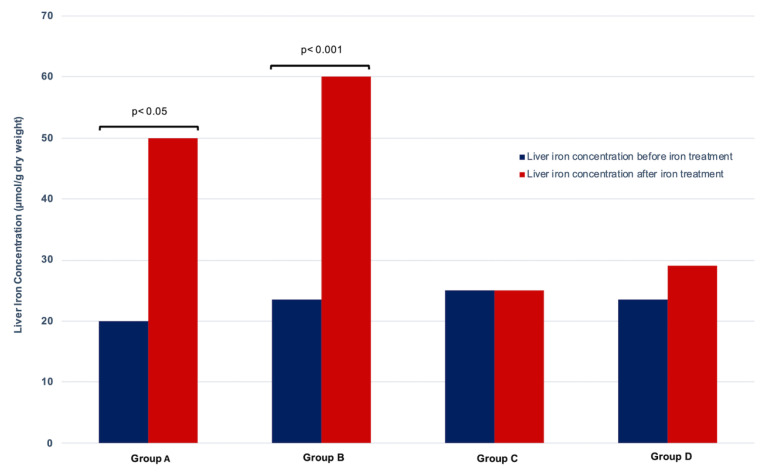
**Evolution of liver iron concentration before and after iron therapy in 54 dialysis patients.** Median liver iron concentration before and after iron therapy in: Group A: 11 patients treated with 1.2 g iron sucrose; Group B: 14 patients treated with 2.4 g iron sucrose; Group C: 15 patients treated with ferric carboxymaltose; Group D: 14 patients treated with iron isomaltoside. *p* value with Wilcoxon test.

**Figure 2 pharmaceutics-14-01408-f002:**
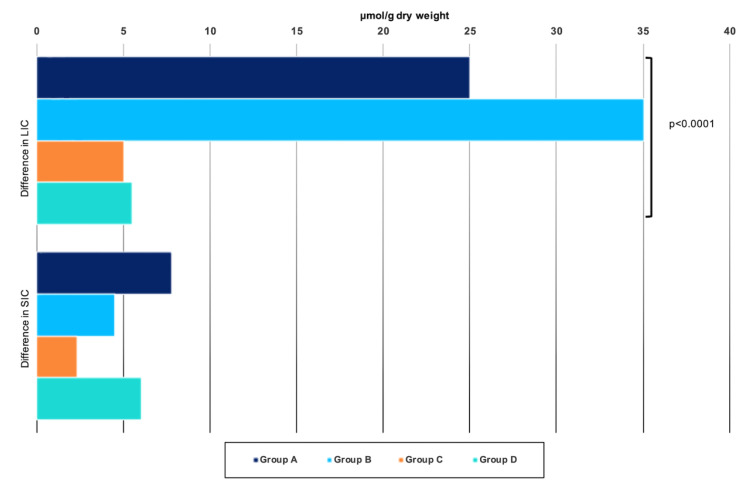
**Absolute difference in liver and spleen iron concentrations in 54 dialysis patients.** Median absolute difference in liver (LIC) and spleen (SIC) iron concentrations in: Group A: 11 patients treated with 1.2 g iron sucrose; Group B: 14 patients treated with 2.4 g iron sucrose; Group C: 15 patients treated with ferric carboxymaltose; Group D: 14 patients treated with iron isomaltoside. *p* value with Kruskal–Wallis test.

**Figure 3 pharmaceutics-14-01408-f003:**
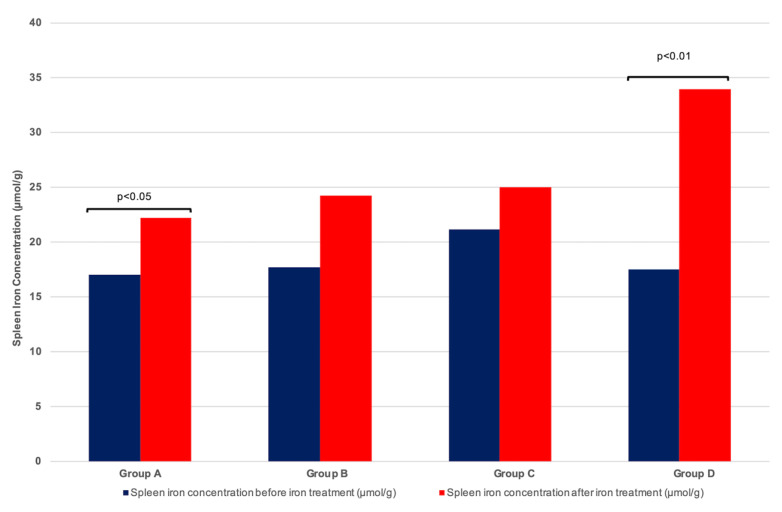
**Evolution of spleen iron concentration (SIC) before and after iron therapy in 54 dialysis patients.** Median SIC before and after treatment in: Group A: 11 patients treated with 1.2 g iron sucrose; Group B: 14 patients treated with 2.4 g iron sucrose; Group C: 15 patients treated with ferric carboxymaltose; Group D: 14 patients treated with iron isomaltoside. *p* value with Wilcoxon test.

**Figure 4 pharmaceutics-14-01408-f004:**
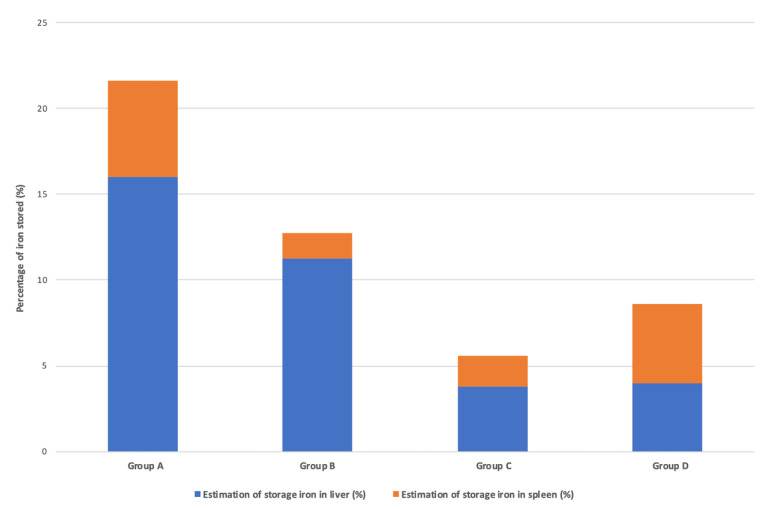
**Percentage of infused iron stored in the liver and spleen in 40 dialysis patients.** Percentage of infused iron stored in the liver and spleen estimated by the Brissot regression slope [32] in: Group A: 11 patients treated with 1.2 g iron sucrose; Group B: 14 patients treated with 2.4 g iron sucrose; Group C: 15 patients treated with ferric carboxymaltose; Group D: 14 patients treated with iron isomaltoside. Estimation of LIC + SIC, *p* = 0.0012 (Chi^2^ test); estimation of LIC, *p* = 0.0015 (Chi^2^ test).

**Table 1 pharmaceutics-14-01408-t001:** Demographic and clinical characteristics of the study population.

	Iron Sucrose 1.2 g	Iron Sucrose 2.4 g	Ferric Carboxymaltose 1 g	Iron Isomaltoside 1 g	
	(*n* = 11)	(*n* = 14)	(*n* = 15)	(*n* = 14)	*p*-Value
	Group A	Group B	Group C	Group D	
**Age (years)**	68 (51–89)	52.5 (30–82)	70 (40–89)	56.5 (33–87)	0.08 *
**Sex, female**	6 (54.5%)	8 (57.1%)	10 (66.7%)	3 (21.4%)	0.09 **
**Dialysis duration before**	6.6 (1.1–48)	13.5 (1.9–139.6)	4.3 (0.7–116.8)	2.25 (0.9–51.8)	0.07 *
**the study (months)**					
**Dialysis modality (HD, PD)**	11 (100% HD)	14 (100% HD)	14 (93.3% HD)	8 (57.1% HD)	
			1 (6.7% PD)	6 (42.9% PD)	
**Diabetes mellitus**	4 (36.4%)	5 (35.7%)	6 (40%)	2 (14.3%)	0.45 **
**Modified Charlson**	6 (3–10)	4.5 (2–8)	6 (2–8)	3.5 (2–11)	0.14 *
**comorbidity index**					
**Audit questionnaire on**	2 (1–5)	1 (0–12)	1 (0–4)	1 (0–10)	0.37 *
**alcoholism**					
**Weight (kg)**	65 (43–97)	73.25 (55–95)	72.5 (52–112)	72.25 (58.5–106)	0.70 *
**BMI (kg/m^2^)**	24 (17–34)	25.5 (19–35)	26 (18–40)	26 (21–35)	0.73 *
**IV iron dose received**	1.2 (1–1.6)	2.4 (2.3–2.7)	1 (1–1.1)	1 (1–1.2)	<0.0001 *
**between the two MRI (g)**					
**ESA therapy (closest to MRI 1)**	10 (90.9%)	13 (92.9%)	13 (86.7%)	10 (71.4%)	0.38 **
**ESA therapy (closest to MRI 2)**	10 (90.9%)	12 (85.7%)	12 (80%)	10 (71.4%)	0.62 **
**Darbopoetin dose**	40 (0–100)	40 (0–100)	40 (0–130)	20 (0–130)	0.06 *
**(μg/week closest to MRI 1)**					
**Darbopoetin dose**	40 (0–100)	40 (0–100)	50 (0–130)	25 (0–130)	0.57 *
**(μg/week closest to MRI 2)**					

Data shown are median (range), or *n* (%). * Kruskal–Wallis test followed by Dunn’s test; ** Chi^2^ test. HD: hemodialysis, PD: peritoneal dialysis, BMI: body mass index, MRI: magnetic resonance imaging, ESA: erythropoiesis-stimulating agent. Reference ranges: Audit questionnaire on alcoholism: <8; BMI: 18.5–25.

**Table 2 pharmaceutics-14-01408-t002:** Biological variables in the four treatment groups before and after IV iron therapy.

	**Group A**				**Group B**			
	**(Changes in 11 patients treated with 1.2 g iron sucrose)**				**(Changes in 14 patients treated with 2.4 g iron sucrose)**			
	**Initial**	**Final**	**Difference**	** *p* ** **-Value**	**Initial**	**Final**	**Difference**	** *p* ** **-Value**
			**[95%CI]**				**[95%CI]**	
**LIC (μmol/g)**	20 (5–50)	50 (30–170)	25 [15–60]	0.001	23.5 (5–45)	60 (38–210)	35 [30–60]	0.0001
**SIC (μmol/g)**	17 (9.6–30.2)	22.2 (11.9–91.4)	7.8 [–5.5–44.1]	0.049	17.7 (11.2–31)	24.3 (12.3–67.7)	4.5 [–4.3–42.2]	0.07
**Spleen R2* (s^−1^)**	31.2 (17.9–54.9)	40.7 (22–163.9)	14 [–9.9–79.1]	0.049	32.5 (20.7–56.5)	44.4 (22.6–122)	8.2 [–7.7–75.8]	0.07
**Hemoglobin**	10.8 (7.2–13.3)	10.5 (8.9–12.2)	0.05 [–0.9–1.5]	0.64	8.85 (6.3–12)	11.8 (9–13.9)	2.2 [1.3–4.3]	0.001
**(g/dL)**								
**CHr (pg)**	28 (24.9–30.9)	29.6 (24.6–32.5)	1.95 [–1.3–4.9]	0.11	27.7 (20.3–32.8)	31 (27.3–35.4)	5.5 [–5.5–8.8]	0.25
**Serum ferritin**	46 (8–112)	56 (12–220)	28 [–6–94]	0.05	21.5 (6–221)	80 (16–443)	47 [9–311]	0.008
**(ng/mL)**								0.004
**Serum iron**	7.8 (2.8–19)	8.8 (3.5–33)	1.4 [0.7–12.6]	0.049	5.3 (2.7–9.5)	9.7 (4.8–20.9)	2.3 [0.6–11.4]	
**(μmol/L)**								
**Serum**	2.3 (1.7–3)	2 (1.5–3.1)	–0.5 [–0.7–0]	0.047	2.6 (1.6–3.2)	2.1 (1.8–2.6)	–0.45 [–0.8–0]	0.016
**transferrin (g/L)**								
**TSAT (%)**	11.56 (4.15–38)	17 (4.52–67.73)	5.44 [0.37–28]	0.02	8.59 (3.38–17.27)	17.08 (10.67–34.83)	5.39 [2.52–27.12]	0.008
**CRP (mg/L)**	3.1 (1–14.8)	1.3 (1–8.4)	–1 [–2.7–1.5]	0.3	2.2 (1–15.9)	3.7 (1–24.2)	0 [–2.6–11.9]	0.31
	**Group C**				**Group D**			
	**(Changes in 15 patients treated with 1 g ferric carboxymaltose)**				**(Changes in 14 patients treated with 1 g iron isomaltoside)**			
	**Initial**	**Final**	**Difference**	** *p* ** **-Value**	**Initial**	**Final**	**Difference**	***p*-Value**
			**[95%CI]**				**[95%CI]**	
**LIC (μmol/g)**	25 (5–69)	25 (5–73)	5 [2–9]	0.07	23.5 (19–41)	29 (20–42)	5.5 [–6–9]	0.14
**SIC (μmol/g)**	21.2 (8.9–58.3)	25 (9.9–103.4)	2.3 [–3.7–21.9]	0.17	17.5 (8.8–87.1)	34 (12.3–79.5)	6 [2.4–15.5]	0.007
**Spleen R2* (s^−1^)**	38.8 (16.6–105.3)	45.7 (18.3–185.2)	4.1 [–6.7–38.9]	0.17	32.2 (16.4–156.3)	61.7 (22.7–142.9)	10.8 [4.2–28]	0.007
**Hemoglobin**	10.1 (7.3–11.7)	11.3 (9–14.5)	1.4 [–0.3–3.5]	0.042	9.8 (7.1–13.5)	11.2 (9.1–15)	1 [0.2–2.5]	0.002
**(g/dL)**								
**CHr (pg)**	29.4 (26.4–34.4)	30.7 (27.9–33.8)	1 [–1.5–3.6]	0.24	32.3 (22.7–35.4)	33.8 (26.3–35.7)	1.55 [–0.2–3.1]	0.11
**Serum ferritin**	30 (15–195)	59.5 (22–644)	12 [–2–177]	0.027	113 (19–363)	260 (38–472)	61 [30–227]	0.008
**(ng/mL)**								
**Serum iron**	8.1 (4.5–11.9)	8.3 (6–16.3)	0.7 [–2.4–6.1]	0.38	10.2 (5.1–18.6)	13.8 (4.5–17.4)	0.2 [–2.5–4.30]	0.61
**(μmol/L)**								
**Serum**	2.2 (1.42–2.9)	2.03 (1.4–2.8)	–0.29 [–0.5–0.1]	0.02	2.25 (1.8–3)	2.05 (1.6–2.8)	–0.20 [–0.32–0.10]	0.1
**transferrin (g/L)**								
**TSAT (%)**	14.73 (6.92–28)	15.88 (9.29–43.50)	3.87 [–2.73–15.50]	0.16	18.62 (7.47–39.16)	26.66 (7.29–40)	2.8 [–2–14.05]	0.09
**CRP (mg/L)**	2.8 (1–11.6)	2 (1–6.1)	0 [–0.5–2.1]	0.22	2.2 (1–27.6)	3.1 (1–24.8)	0.60 [–7.9–3.4]	0.72

Data shown are median (range); *p* value determined using the Wilcoxon paired test. LIC: liver iron concentration, SIC: spleen iron concentration, CHr: hemoglobin reticulocyte count, TSAT: transferrin saturation, CRP: C-reactive protein, CI: confidence interval. Reference ranges: Hemoglobin: 12.9–16.7 g/dL (male), 11.5–15.1 g/dL (female); CHr: 32.1–38.8 pg; serum ferritin: 22–275 ng/mL (male), 15–204 ng/mL (female); serum iron: 11.6–31.3 μmol/L (male), 9–30.4 μmol/L (female); serum transferrin: 1.63–3.44 g/L (male), 1.73–3.60 g/L (female); TSAT: 20–40% (male), 15–35% (female).

## Data Availability

The deidentified and anonymized data will be made available with publication upon reasonable request. Proposals should be directed to rostotom@orange.fr.

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
