# Peer review of "Differential Pharmacokinetics of Liver Tropism for Iron Sucrose, Ferric Carboxymaltose, and Iron Isomaltoside: A Clue to Their Safety for Dialysis Patients"

_pharmaceutics, 2022, doi:10.3390/pharmaceutics14071408_

Round 1

Reviewer 1 Report

Thank you for the opportunity to review the manuscript. 

Please see below for comments: 

Line 43 – spell out HLA on first mention.

Lines 47 – 48, is it only “functional” iron deficiency vs IDA.

Line 49, IV superior to oral preparations for HD or PD?

Line 64, …at least the next decade….how does one know this timeline?

Lines 73-75, not sure if this belongs here? Perhaps, I am not fully understand what it is mean in the context…”demonstrate”?

Line 79, the study started in 2005? This was a very long time ago; but then later on in the manuscript, it stated a different timeline in lines 189 to 190. Not clear?

Line 100, the last sentence….”…..in some.”  - what is “some” refer to?

Results section, starting in line 189…..I am a little unsure…the study started in 2005 (now we are in 2022, it’s almost 20 yrs), but the numbers of patients is very small 54.

Reviewer 2 Report

The authors investigate hepatic iron overload in a clinical study on dialysis patients receiving anemia treatment, including erythropoiesis-stimulating agents and intravenous iron. Patients received one of four iron treatments: low dose iron sucrose, high dose iron sucrose, ferric carboxymaltose, or iron isomaltoside. Hepatic iron overload was assessed by MRI. The authors found increasing hepatic iron in the groups treated with iron sucrose and conclude that the alternative iron treatments should be preferred.

The study question is relevant, the data are interesting, and the paper is generally well-written. However, I have major concerns about the presentation of the study design and the conclusions drawn from the findings.

+++ major +++

-        What kind of study is this? Obviously this is not an RCT, but what is a “cross-sectional and longitudinal” study? Please add a chart about the timeline of patient enrollment, examination and treatment time points. The description is confusing: The study is both cross-sectional and longitudinal, has been started in 2005, but performed between August 7, 2013 and January 14, 2020?

-        Crucial: Since there was no randomization, patients allocated to the different treatmens do have very different characteristics, comorbidities and thus treatment responses. It is therefore not permitted to assign such strong effects to the treatment alone. Since the sample size does not allow for a proper statistical model adjusting for the differences in characteristics, the analysis is constrained to groupwise comparisons. This the major limitation of the study and should be thoroughly discussed in the discussion section. Furthermore, due to this major limitation, the language about the superiority of alternative iron regimen and “abandonment of sucrose” must be massively toned down, because such strong conclusions cannot be drawn from the presented data.

-        Group D: There should be a discussion why spleen iron is increasing so much.

-        Please remove all 3D effects from the figures and present regular bar charts.

-        Why was the analysis on sequestered iron not done for group B?

-        Systemic inflammation seems to have developed during treatment in groups B and D, any explanation why?

-        The transformation of relaxation time to mumol/g required quite a lot of assumptions and extrapolations. Raw units, i.e. R* in s-1 should be given as well. This would also simplify comparison with other studies.

  +++ minor +++

-        Results and conclusions should not be presented at the end of the introduction, the last paragraph should thus be omitted.

-        Initial LIC and SIC are duplicated between table 1 and 2 (by the way, typo in initial SIC in group D)

-        Line 170: “Quantitative data are given as number and percentage “ – do you mean categorical data, as opposed to continuous? Because both are quantitative.

Round 2

Reviewer 1 Report

Thank you for the opportunity to re-review the manuscript.

I think the authors adequately addressed prior comments and the manuscript is now improved. 

Reviewer 2 Report

I thank the authors for providing a revised version of their manuscript. Importantly, limitations of the study are now transparently discussed. I have no further remarks.